# Spectral Characteristics and Sensor Ability of a New 1,8-Naphthalimide and Its Copolymer with Styrene

**DOI:** 10.3390/s20123501

**Published:** 2020-06-21

**Authors:** Desislava Staneva, Silvia Angelova, Ivo Grabchev

**Affiliations:** 1Department of Textile and Leather, University of Chemical Technology and Metallurgy, 1756 Sofia, Bulgaria; grabcheva@mail.bg; 2Institute of Optical Materials and Technologies “Acad. Jordan Malinowski”, Bulgarian Academy of Sciences, 1113 Sofia, Bulgaria; sea@iomt.bas.bg; 3Faculty of Medicine, Sofia University “St. Kliment Ohridski”, 1407 Sofia, Bulgaria

**Keywords:** 1,8-naphthalimide, polystyrene, fluorescent sensor, photo-induced electron transfer, metal ions

## Abstract

In this study, a novel 6-(allylamino)-2-(2-(dimethylamino)ethyl)-1H-benzo[de]isoquinoline-1,3(2H)-dione (NI3) was synthesized and characterized. Its copolymer with styrene was also obtained. The photophysical characteristics of NI3 were investigated in organic solvents and the results were compared with those of its structural analogue, 2-allyl-6-((2-(dimethylamino)ethyl)amino)-1H-benzo[de]isoquinoline-1,3(2H)-dione (NI4). The influences of the pH in the medium and different metal ions on the fluorescent intensity of monomers and polymers were also investigated. Computational tools (DFT and TDDFT calculations) were employed when studying the structure and properties of the 1,8-naphthalimide-based chromophores. Although the position of the *N,N*-dimethylaminoethylamine receptor fragment did not significantly impact proton detection, it was still important for detecting metal ion sensor ability, especially for monomeric 1,8-naphthalimide structures and their copolymers with styrene.

## 1. Introduction

In recent years, 1,8-Naphthalimide derivatives have become one of the most investigated fluorescent chromophore systems in different vanguard areas. The relatively low molecular weight, compact molecule, and the possibility to introduce active groups capable of participating in polymerization processes makes these compounds suitable for structural modification of polymer materials with fluorescent properties [1,2], or as free-radical polymerization photoinitiators upon visible light exposure [3,4]. Moreover, researchers have found different biological and medical applications as antibacterial, antifungal, anticancer, and antiviral agents [5,6,7,8,9,10,11].

Derivatives of 1,8-naphthalimide have been increasingly investigated as a signaling unit in the design of sensor systems for the detection of cations, anions, or neutral analytes, which are major industrial environmental pollutants [12,13,14,15,16,17]. For the first time, we have combined the sensor activity of 1,8-naphthalimides with their ability to copolymerize with vinyl and methacrylate monomers to produce polymeric sensors [18]. An allyl group directly attached to the imide nitrogen atom was used as a polymerization group. On the other hand, the groups as *N,N*-dimethylaminoethylamino [18], *N,N*-dimethylaminoethyloxy [19], or *N*-methylpiperazine [20] were introduced at the C-4 position for their use as receptor fragments to react with metal ions and protons in the sensor system design, working in the photo-induced electron transfer (PET) mode.

In this work, a novel 1,8-naphthalimide derivative and its copolymer with styrene was synthesized and characterized with regard to investigate their structure and spectral characteristics. The detection ability of both compounds was investigated. Results found that the polymer sensor can be repeatedly used as a heterogeneous fluorescent sensor for detection of Fe(III) in aqueous solutions.

## 2. Materials and Methods

### 2.1. Synthesis

Synthesis of 2-(2-(dimethylamino)ethyl)-6-nitro-1H-benzo[de]isoquinoline-1,3(2H)-dione (NI2) occurred as follows: 2.43 g (0.01 mol) of 4-nitronaphthalic anhydride was dissolved in 50 mol ethanol and 1 mL (0.01 mol) mol *N,N*-dimethylaminoethylendiamine was added drop wise to the solution and mixture. It was then sired for 3 h at 70 °C. The process was controlled by thin-layer chromatography and the final product was filtered off with a very high yield and purity after pouring the liquor into 500 mL of water. Yield: 94.2%; FT-IR (KBr) cm-1: 3077, 2942, 2820, 2774, 1704, 1655, 1623, 1732, 1336, 1242, 1161, 1046, 836, 786 and 759.

^1^H NMR (CDCl3, δ, ppm): 8.78 (d; J = 7.14 Hz; 1H); 8.41 (d; J = 8.01 Hz; 1H); 8.36 (d; J = 8.12 Hz, 1H); 8.24 (dd, J = 7.31 Hz; 1H); 7.94 (d; J = 8.10 Hz; 1H); 4.08 (t; J = 6.86; 2H); 2.62 (t; J = 6.91 Hz; 2H); 2.14 (s; 6H); ^13^C NMR (CDCl3, δ, ppm): 165.2, 164.4, 155.2; 131.2, 131.5, 129.0, 125.2, 123.1, 120.3, 119.5, 109.3, 101.4, 59.5, 49.1 and 45.1.

The elemental analysis occurred as follows: C_16_H_15_N_3_O_4_ (313.11 g mol^−1^): Calc. (%): C 61.32, H 4.79, N 13.41; found (%): C 61.43, H 4.73, N 13.39.

Synthesis of 6-(allylamino)-2-(2-(dimethylamino)ethyl)-1H-benzo[de]isoquinoline-1,3(2H)-dione (NI3) occurred as follows: 3.13 g (0.01 mol) of compound NI2 was dissolved in 20 mL of *N,N*-Dimethylformamide (DMF) and 1.1 mL (0.01 mol) allylamine was added. The mixture was left in the dark for 24 h then poured into 100 mL of a water-ice mixture. The precipitate was filtered, washed with water, and dried under a vacuum at 40 °C. Yield was 93%; FT-IR (KBr) cm^−1^ was 3091, 2965, 2891, 2785, 1673, 1640, 1583, 1465, 1388, 1341, 1244, 1123, 1017, 773 and 754. Next, ^1^H NMR (CDCl3, δ, ppm): 8.47 (d; J = 7.29 Hz; 1H); 8.38 (d; J = 8.42 Hz; 1H); 8.06 (d; J = 8.45 Hz, 1H); 7.53 (dd, J = 7.6 Hz; 1H); 6.6 (d, J = 8.49 Hz; 1H); 6.01 (m; 1H); 5.69 (t; J = 5.38 Hz; 1H); 5.35 (dd; J = 17.35 Hz; J = 1.26 Hz; 1H); 5.28 (dd; J = 10.34; J = 1.24 Hz; 1H); 4.33 (2H; J = 6.86; 2H); 4.03 (t; J = 5.34 Hz; 2H); 2.72 (t; J = 6.8 Hz; 2H); 2.41 (s; 6H); ^13^C NMR (CDCl3, δ, ppm): 164.7, 164.1, 143.3, 134.3, 133.0, 131.1, 129.6, 126.0, 124.6, 122.8, 120.1, 117.6, 110.2, 104.7, 57.1, 45.0, 45.7, 37.8; elemental analysis: C_19_H_20_N_3_O_2_ (322.18 g mol^−1^): Calc. (%): C 70.56, H 6.20, N 13.03; found (%): C 70.43, H 6.13, N 13.09.

Synthesis and characterization of poly(St-co-NI3) occurred as follows: Fluorescent poly(St-co-NI3) was synthesized by radical co-polymerization of styrene with NI3 (1.0 wt%) at 70 °C for 8 h and 1.0 wt% dibenzoylperoxide as an initiator. The transparent poly(St-co-NI3) was precipitated with ethanol from toluene several times, washed with ethanol, and dried. All photophysical and spectral and sensor measurements were carried out with precipitated polymers.

### 2.2. Analysis

UV-Vis spectrophotometric investigations were performed on a Thermo Spectronic Unicam UV 500 spectrophotometer. Fluorescence spectra were taken on a “Cary Eclipse” spectrofluorometer. All spectra were recorded using a 1 cm path length synthetic quartz glass cell at a 10^−5^ mol L^−1^ concentration. The fluorescence quantum yield was determined on the basis of the absorption and fluorescence spectra. Rhodamine (Φ_ref_ = 0.94) was used as a reference [21]. The effect of the metal cations and protons on the fluorescent intensity was examined by adding a few μL of the metal cations stock to a known volume of the sensor solution (3 mL) [22]. The reproducibility of the results was 99%. ^1^H NMR (600.13 MHz) and ^13^C (150.92 MHz) spectra were acquired on an AVANCE AV600 II+ NMR spectrometer. Fourier-transform spectrometer (IRAffinity-1 Shimadzu, Kyoto, Japan) with the diffuse-reflectance attachment (miracle attenuated total reflectance attachment) occurred at a 2 cm^−1^ resolution.

Thin layer chromatographic (TLC) analysis was followed by silica gel (Fluka F60 254 20 × 20; 0.2 mm) using an n-heptane/acetone (1:1) solvent system as an eluent. Next, 4-Nitro-1,8-naphthalic anhydride, ethylenediaminetetraacetic acid (EDTA), *N,N*-dimethylaminoethylendiamine, allylamine, CuSO_4_·5H_2_O, CoSO_4_·7H_2_O, AgNO_2_, ZnSO_4_·H_2_O, and Ni(NO_3_)_2_·6H_2_O were obtained using Sigma-Aldrich (Munich, Germany). The molecular weight characteristics of poly(St-co-NI3) were determined on a gel permeation chromatography (GPC) water 244 apparatus equipped with combinations of 100 and 1000 A, linear Ultrastyragel columns. The solvent was THF at a flow rate of 1.0 mL min^−1^ at 450 °C. Both the differential refractive index and UV-vis absorption detector (λ_A_ = 423 nm) were used. Polystyrene calibration was used for molecular weight calculations.

### 2.3. Computational Details

Molecular ground state geometries for NI3 and NI4 products were fully optimized using B3LYP [23,24] functional. The diffuse function-augmented version of Pople’s basis set 6–31G(d,p) [25,26,27] was used. The default optimization algorithm included in the Gaussian 09 [28] and convergence criteria were applied. C1 symmetry (i.e., no symmetry) was assumed for all systems under study. Local minima were verified by establishing that the Hessians had only positive eigenvalues. The B3LYP/6–31+G(d,p) optimized geometries in the gas-phase were re-optimized in toluene, methanol, and water by using the IEFPCM (Integral Equation Formalism Polarizable Continuum Model) -[29] method. The solvents were defined by the Gaussian 09 default internal parameters, including the dielectric constant ε values. TDDFT (Time-Dependent Density Functional Theory) excitation energies were computed at the TDPBE0/6-311+G(2d,p)//B3LYP/6-31+G(d,p) level; in each TDDFT calculation, the lowest 20 excited states were computed. Solvent effects were included in TDDFT calculations (via IEFPCM). The B3LYP functional and 6-31+G(d,p) basis set was employed in optimizing the structures of the NI-metal complexes, and evaluating the respective Gibbs free energies. The differences ΔE_el_, ΔE_th_, PΔV (work term), and ΔS between the products of the complex formation reactions (complexes) and reactants (NI ligands and metal cations) were used to evaluate the Gibbs energy of the complex formation in water, ΔG^78^, at T = 298.15 K according to the equation:ΔG^78^ = ΔE_e_l + ΔE_th_ + PΔV − TΔS(1)

A negative ∆G implies a thermodynamically favorable complex formation, where a positive value implies an unfavorable one. The PyMOL molecular graphics system was used to generate the molecular graphics images [30].

## 3. Results

### 3.1. Synthesis of 6-(Allylamino)-2-(2-(Dimethylamino)Ethyl)-1H-Benzo[de]isoquinoline-1,3(2H)-Dione (NI3)

Herein, 4-Nitro-1,8-naphthalic anhydride (NA1) was used as an initial product for the synthesis of NI3 (Scheme 1). Its reaction with *N*, *N*-dimethylaminoethylendiamine in an ethanol solution yielded 2-(2-(dimethylamino)ethyl)-6-nitro-1H-benzo[*de*]isoquinoline-1,3(2H)-dione (NI2). The final product (NI3) was obtained after nucleophilic substitution of the nitro group from NI2 with an allylamino group. The chemical structure of the new compound NI3 was characterized by FTIR, ^1^H and ^13^C NMR spectra, as well as an elemental analysis.

The chemical structure of 2-allyl-6-((2-(dimethylamino)ethyl)amino)-1H-benzo[de]isoquinoline-1,3(2H)-dione (NI4) is shown in Scheme 2. Its synthesis was previously described [18]. When comparing its chemical structure with the new compound NI3, the difference between them was only in the position of the allyl group and the *N*,*N*-dimethylaminoethylendiamine receptor fragment. This enabled us to investigate the effect of the receptor fragment on the photophysical characteristics and sensor activity of both 1,8-naphthalimides, when it was attached to the imide nitrogen atom (NI3) or to the C-4 nitrogen atom of the 1,8-naphthalimide chromophore system (NI4). Both compounds were used as fluorescent monomers for the structural modification of polystyrene.

The molecular weight characteristics obtained from GPC (Mw and Mn) and polydispersity Mw/Mn of poly (St-co-NI3) and poly(St-co-NI4) are summarized in Table 1. The data confirms the formation of high molecular weight polymers. Comparing the data for both copolymers, it is seen that poly(St-co-NI4) has higher molecular weights due to the position of the allyl group. In this case, the allyl group bonded to the amide nitrogen atom was more polarized and more active than when it was at the C-4 position. This, in turn, led to the production of a copolymer with a higher molecular weight. Due to the low concentration of the NI3 linked to the main polystyrene macromolecule, it was not possible to use NMR spectroscopy to unequivocally prove the chemical bond between them. The new poly(St-co-NI3) copolymer was characterized by gel permeation chromatography with double detection: refraction and UV-Vis at 423 nm. The elution time values obtained in both chromatograms were very close (Figure 1), which indicates that poly(St-co-NI3) was absorbed at 423 nm. This result is indicative of formation of covalent bond between NI3 and main polystyrene chain.

The amount of chemically bonded 1,8-naphthalimide fluorophores were determined by comparing the fluorescent intensity of the precipitated and unprecipitated polymer, as well as the use of fluorescence spectroscopy. In this case, fluorescence spectroscopy was preferable to the absorption as a more sensitive analytical method. Covalently bonded to the polymer chain 1,8-naphthalimide, fluorophores were found at a concentration of 0.75% for NI3 and 0.94% for NI4. There was no significant effect on the molecular and film-forming characteristics of the copolymers (Table 1). On the other hand, in this concentration range, both 1,8-naphthalimides were sufficient to exhibit their sensor qualities.

### 3.2. Spectral Characteristics of NI3 and NI4

The photophysical characteristics of 1,8-naphthalimide derivatives depended on the polarization of 1,8-naphthalimide chromophore systems due to the electron donor-acceptor interaction occurring between the carbonyl groups from the imide structure and the substituents at the C-4 position.

The absorption and fluorescence spectral bands of the 1,8-naphthalimide derivatives largely depended on the electron donating power of the substituents at the C-4 position. Figure 2 plots an example of the absorption and fluorescence spectral profile of NI in an acetonitrile solution. In the case of 4-substituted-1,8-naphthalimide derivatives, a donor-acceptor interaction took place between the substituent at the C-4 position and carbonyl groups, which caused a band formation in the absorption spectra with a charge transfer character. The figure also shows that the absorption and fluorescence spectra were mirror images with a small overlap area, which is an indication that no aggregation processes occurred at this concentration. Similar spectral curves were observed in all other studied organic solvents.

The absorption and fluorescence spectral bands of the 1,8-naphthalimide derivatives largely depended on the electron donating power of the substituents at the C-4 position. On the other hand, the organic solvents played a significant role in the photophysical properties via their polarity and possibility to participate in intermolecular interactions with the 1,8-naphthalimide chromophore systems [31,32]. The effect of solvent polarity on the photophysical properties of NI3 was investigated in 10 organic solvents with different polarities: the absorption (λ_A_) and fluorescence (λ_F_) maxima, the extinction coefficient (ε), the Stokes shift (ν_A_−ν_F_), and the quantum yield of fluorescence (Φ_F_) are summarized in Table 2.

In all investigated organic solvents, NI3 had a yellow color with yellow-green fluorescence. The long-wavelength bands of the absorption spectra in the visible region were bands of charge transfer/CT/due to the π→π electron transfer on the S_0_→S_1_ transition. The absorption maxima λ_A_ were in the visible region at 414–436 nm and the respective fluorescence maxima were between 478 ÷ 529 nm. Figure 3 presents the dependence of the absorption position and fluorescence maxima of compound NI3 from the polarity of organic solvents. As seen, the nature and the polarity of the organic solvents had significant influence on the spectral properties of NI3. Changes in the position of the NI3 CT bands in the solvents were brought about by the solvents’ solvatochromic effect. In comparison with the polar methanol, the absorption and fluorescent maxima in apolar toluene solvent were hypsohromically shifted (Δλ_A_ = 22 nm and Δλ_F_ = 51 nm), as the polarization of the chromophoric molecules depended strongly on the solvents polarity, as well as on the specific NI3-solvent interactions, which caused a change in the polarization of the 1,8-naphthalimide chromophore system. As can be seen, NI3 displayed a positive solvatochromism.

The Stokes shift (υ_A_−υ_F_) is a parameter that indicates the difference in the properties and structure of the chromophores between the ground state S_0_ and the first exited state S_1_. The obtained values for NI3 were between 3234–4084 cm^−1^, which were similar with other 1,8-naphthalimides tat had secondary amino groups as substituents at the C-4 position [1,33]. From the data in Table 1 and Figure 1, it can be also seen then the values of the Stokes shift depended on the nature of organic solvents, which was higher in the case of polar solvents.

The ability of NI3 to emit absorbed light energy was characterized by the quantum yield of fluorescence Φ_F_. It was determined on the basis of respective absorption and fluorescence spectra. As can be seen from the data in Table 1, the quantum yields of NI3 were between Φ_F_ = 0.53–0.71. These results were similar to other 1,8-naphthalimides with secondary amino groups as substituents at C-4 [1,34,35].

Comparing the photophysical characteristics of NI3 with those obtained for NI4 [36], it can be seen that the position of the substituents had an analogous effect on their spectral characteristics. In both compounds, the substituents at the C-4 position were with secondary amino groups (allylamino for NI3 and *N,N*-dimethylaminoethylamine for NI4) and the polarization of the chromophore system was approximately the same. A significant difference was found when examining the quantum yield of the compounds (Figure 4).

For NI3, the values obtained were in the range 0.53–0.71, which was a good emission in every tested solvent. The polarity of the medium did not significantly affect the emitted fluorescent intensity (Figure 3). NI4 showed a huge difference in quantum efficiency in organic solvents, with a strict dependence on polarity. In nonpolar solvents, the quantum yield of NI4 was very high (Φ_F_ = 0.88 at chloroform) and the fluorescence was quenched significantly in polar solvents (Φ_F_ = 0.008 at methanol) [18]. The difference of the fluorescent intensity was explained by a possible PET process, which was carried out in polar media and accompanied by weak fluorescence emission and quenched in nonpolar media, whereby the fluorescence emission was recovered. Brawn et al. also noted this behavior, investigating similar 1,8-naphthalimide derivatives in organic solvents [37].

To understand the molecular level, the relationship between molecular composition, electronic characteristics, and photophysical behavior, we theoretically investigated NI3 and NI4 via popular and affordable DFT calculations. As a first step, a full geometry optimization of the molecular structures was performed at B3LYP/6-31+G(d,p) level of theory in the gas-phase and in toluene (ε = 2.4) and methanol (ε = 32.6) solvent environments. The gas-phase optimized low-energy isomers of NI3 and NI4, which are visualized in Figure 5. TDDFT calculations at TDPBE0/6–311+G(2d,p)//B3LYP/6-31+G(d,p) level were used to probe the electronic reorganization upon excitation of the studied molecules in toluene and methanol.

According to the experimental results, the nature and the polarity of the organic solvents significantly influenced the spectral properties of NI3. The computational approach used to account for the solvent effects in the studied solvatochromic probes was an implicit solvent model (PCM). A well-known significant drawback of implicit approaches is that they do not account for specific solute-solvent interactions and fail to reproduce the experimental sequence of solvatochromic shifts that correlate with static dielectrics constant with the solvent [38]. The data reported in Table 3 indicates that the TDDFT method with a hybrid XC functional (PBE0) is suitable to reproduce the linear optical properties of compounds NI3 (and NI4), including the contrast between the absorption spectra of the compounds in a non-polar (toluene) and a polar (methanol) medium. A typical systematic overestimation of the excitation energies (in comparison to the experimental values) by the TDDFT schemes [39] was observed for NI3: 0.07 eV in toluene and 0.12 eV in methanol.

The first excited states were determined by HOMO-LUMO transitions with oscillator strengths *f* in the range 0.27–0.34. The optimized geometries and frontier orbitals of compounds NI3 and NI4 are shown in Figure 5. HOMO orbitals were located differently for NI3 and NI4. For NI3, it was located on the *N*,*N*-dimethylaminoethylendiamine receptor fragment and for NI4 it was located entirely on the 1,8-naphtalimide core. LUMOs were delocalized on the 1,8-naphtalimide cores for both compounds. These differences in the spatial distribution of the frontier orbitals (HOMOs) explain the different photophysical properties of NI3 and NI4.

Lippert-Mataga investigated the solvent polarity of NI3 and the change of the dipole moment upon excitation, as well as the relation between the dependence of the Stokes shift and ∆f, as plotted in Figure 6. The radius of the Onsager cavity (a_0_ = 5.83 Å) was determined by quantum chemical calculations (the Monte-Carlo method of calculating molar volume). It was taken as the sphere equivalent radius of the volume inside a contour of 0.001 electrons, per Bohr^3^ density, plus 0.5 Å (recommended a_0_ value for Self-Consistent Reaction Field calculations). The solvents diethyl ether, tetrahydrofuran, and chloroform were excluded. The slope of the line yielded the dipole moment (Δμ = μ_e_ − μ_g_) as a difference between the excited state S_1_ and ground state S_0_ dipole moments, μ_e_ and μ_g_, respectively. The DFT calculated the dipole moment in the ground state μ_g_ to be 9.0 D in toluene and 11.0 D in methanol. Calculated Δμ = μ_e_ − μ_g_ values indicated an increase of the dipole moment upon the transition into the excited state from 9.0 D to 9.8 D in toluene and from 11.0 D to 15.0 D in methanol.

The photophysical characteristics of the poly(St-co-N3) were also examined in the toluene solution, where the copolymer was well soluble and in a thin polymer film (60 μm). In 2% toluene solution, poly(St-co-NI3) has absorption maximum at 408 nm and emits fluorescence with maximum at 474 nm, with a Stokes shift of 3412 cm^−1^. The calculated quantum yield was Φ_F_ = 0.51. These values were very close to those obtained from the free monomeric NI3. This indicates that the chromophore system did not change during the co-polymerization, nor for the main polymer chain binding. Figure 7 shows the excitation and fluorescence spectra of the thin poly(St-co-NI3) film. The excitation spectrum maximum was at 451 nm, while the fluorescence peak was at 502 nm. The bathochromic shift of excitation and fluorescence maxima compared to those in the toluene solution can be explained by the solid fixation of NI3 into the polymer matrix and the inability to undergo conformational changes [40,41].

### 3.3. Influence of pH of the Fluorescent Intensity

The influence of pH of the medium (ethanol and water 1:4, v/v) on the fluorescent intensity of NI3 was investigated. Figure 8 shows the change in fluorescent intensity with the variation of pH. In an alkaline medium (pH = 8–10), the fluorescent intensity was insignificant (ΦF = 0.004). In the pH = 6–8 range, it enhanced more than 100 fold and in the acidic medium, at pH = 3, the quantum yield was ΦF = 0.48. In this case, PET from the tertiary amino group to the chromophore system was carried out. In an acidic environment, the electron transfer was stopped by protonation of the tertiary amino group and the fluorescence was restored (Scheme 3).

The dependence of the fluorescent intensity from pH was analyzed by Equation (2):pH = pKa + log (I_F_ max−I_F_)/(I_F_−I_F_ min)(2)

The calculated pKa value was 7.38.

The Hirshfeld atomic charge was obtained for the optimization in water environment. NI3 geometry showed that *N,N*-dimethylamino group protonation was the most probable one (Figure 9). DFT calculations were carried out on a different *N*-protonated species in the gas-phase and in water. It was further confirmed that the protonation occurred in the *N,N*-dimethylamino group’s nitrogen atom (Table 4).

### 3.4. Influence of Metal Cations on the Fluorescent Intensity of NI3 and NI4

Different metal cations (Cu(II), Fe(III), Pb(II), Zn(II), Co(II) and Ni(II)) were used to investigate the detection ability of NI3 and NI4. As a qualitative indicator of the influence of metal ions on a compound’s fluorescent intensity, the fluorescence enhancement factor (FE = I/Io) was calculated from the ratio of the maximum fluorescent intensity (I: after adding metal ions) and the minimum fluorescent intensity (Io: free of metal ions). Figure 10 plots the enhancement of the fluorescent intensity of both compounds in the presence of the metal cations. An enhancement of the fluorescence intensities was observed in NI4 (FE = 4.5 ÷ 9.2), while in NI3 the metal ions had a negligible effect on fluorescent intensity (FE = 1.2÷1.5). This was true except for Fe(III), where the increase in fluorescent intensity was significant (FE = 6.7 and Zn(II) with FF = 5.2). The great difference in the fluorescent intensity of NI3 and NI4 was induced by metal cations and can be explained by the different structure and stability of the respective tetraaqua complexes.

In the case of NI4, metal ions form a coordinate bond with the free electron pairs of the nitrogen atoms of the C-4 receptor fragment. As a result, the PET from the distant tertiary nitrogen atom was quenched and the fluorescent intensity increased. A similar formation of a metal complex with the substituent at the C-4 position for NI3 was not possible. The respective values for FE were low.

Different possible NI3-Zn^2+^ and NI4-Zn^2+^ complexes were modeled and the Gibbs energies of the complex formation reactions with hydrated Zn^2+^ cations (Zn(H_2_O)_6_^┐2+^) were evaluated in a water environment. To determine the geometries of NI-Zn(II) complexes, hydrated metal cations were placed close to the nitrogen atoms of the ligands and allowed to relax. Figure 11 visualizes the optimized structures of the complexes. The Gibbs free energy for the NI3/ Zn^2+^ metal ion complex formation reaction indicated a spontaneous and energy-favorable complex formation process in water with participation of the imide nitrogen atom (ΔG^78^ = −14.2 kcal mol^−1^). The complex formation with the substituent at the C-4 position was predicted to be an energy-unfavorable process (ΔG^78^ = 18.0 kcal mol^−1^). In the NI3-Zn-(H_2_O)_4_^┐2+^ complex, the Zn^2+^ ion coordinated with the tertiary amino group of the receptor fragment attached to the imide group, as well as indirectly with the carbonyl groups via water-mediated interactions. The initial positions of the water molecules around the metal cation were not preserved. After a reorganization of first-shell water molecules, two of them were displaced and created water bridges with the carbonyl groups. For NI4 the preferred binding position of the metal cation was close to the nitrogen atoms of the C-4 receptor fragment.

Figure 12 shows the fluorescence spectra of NI3 on titration with Fe(III) ions. We observed a significant increase in fluorescence without changing the position of the fluorescence maximum.

A complex between NI3 and Fe(III) ions was also modeled (Figure 13). The value of the Gibbs energy indicated a strong preference towards the triply charged Fe(III) ions (ΔG^78^ = −31.2 kcal mol^−1^). As Figure 11 and Figure 13 demonstrate, the first-shell water molecules in NI3-Zn-(H_2_O)_4_^┐2+^ and NI3-Fe-(H_2_O)_4_^┐3+^ were arranged differently around the metal center. As a result, Fe(III) coordinated strongly with the tertiary amino group of the receptor fragment (2.18 Å bond length) and with one of the carbonyl group (a water mediated contact).

The fluorescent intensity of NI3 depended linearly on the concentration of Fe (III) ions in the range 0 ÷ 1 × 10^−5^ mol L^−1^. Increasing the concentration to 3 × 10^−5^, the intensity slightly decreased (Figure 14A). A very good linear dependence (R = 0.9953) was obtained, which made it possible to calculate, via linear regression [42], the LOD detection limit (5.1 × 10^−6^ mol L^−1^) as well as the limit of quantization (LOQ = 1.70 × 10^−5^ mol L^−1^).

The association constant *Ka* of Fe(III) complex with NI3 was calculated from the Benes-Hildebrand plot [43]:(3)$$\frac{1}{F-Fo}=\frac{1}{Ka\;\left(Fmax-Fo\right)\;\left[Fe\left(III\right)\right]}+\frac{1}{Fmax-Fo}$$
where *F* is the measured fluorescent intensity, *Fo* is the fluorescent intensity of free ligand NI3, and *Fmax* is the saturated fluorescent intensity of the complex NI3-Fe(III). From the obtained linear relationship (Figure 14B) (R^2^ = 0.996), the association constant of NI3 with Fe(III) ions was calculated as Ka = 1.12 × 10^5^ M^−1^.

### 3.5. Detection Ability of Copolymers in Solid State

Planning to obtain a heterogeneous fluorescent sensor, the influence of metal cations on the fluorescent intensity of thin film of poly(St-co-NI3) was evaluated in a buffer solution (CH_3_COOH/NaCH_3_COO^−^ at pH = 5) that contained metal cations at a concentration of 10^−5^ mol L^−1^. As can be seen from Figure 10, only in the presence of Fe(III) and Zn(II) did the NI3 fluorescent intensity increase, which can be considered as ion selectivity compared to the other ions studied. This motivated us to study the metal ion detection ability of the copolymers. The rigidity of poly(St-co-NI3) polymer structure of the film hindered the penetration of the metal cations and their contact with the NI3 receptor fragments. The hydrophobic nature of the polystyrene matrix also delayed inward ion penetration and required a longer contact of the polymer with the metal ions. After 20 min of contact time between the polymer matrix and the cations, their effect was monitored by the change of the fluorescent intensity. Figure 15 shows the decrease in the fluorescent intensity of poly(St-co-NI3) and poly(St-co-NI4) in the presence of Fe(III) and Zn(II) ions. Metal ions Cu(II), Pb(II), Co(II), and Ni(II) exhibited similar behaviors as Zn(II) ions.

For Fe(III) cations, the fluorescent intensity of both polymers decreased in the first 10 min and after that, the change was not significant. The quenching of the fluorescent intensity was 15% for (St-co-NI4) and 10% for poly(St-co-NI4). For other ions it was 1–2%. The results show a high selectivity of the copolymers for Fe(III) in the presence of other metal cations under study in the same concentrations. Due to the smaller ionic radii, the Fe(III) cations had a competitive advantage over the rest of cations in the polymer matrix penetration, as well as a higher ability to react with the sensor receptor. The decrease in the fluorescent intensity was likely due to the formation of a complex in which the Fe(III) ions directly coordinated with the receptor fragment’s nitrogen atom and the NI3 carbonyl groups. This led to a polarization change of the chromophore system and the emitted fluorescence decreased [20,44].

The possibility of copolymers to form a thin film and the eliminating the migration ability proves that the presented sensor can have multiple uses for detecting Fe(III) cations in water sources. The polymer film can be regenerated from the trapped Fe(III) ions using a treatment with an aqueous EDTA solution at 40 ºC for 20 min. In this case, the iron ions formed a new complex outside the receptor fragment of 1,8-naphthalimide. Therefore, the fluorescence of the polymer was restored almost quantitatively to the initial state. This is a reliable indicator of the Fe(III) release from the polymer matrix and its suitability for reuse. The process was repeated six times and proved that fluorescence was quenched and restored to almost the same value after each use. This assumes that the 1,8-naphthalimide sensor fragments do not migrate from the polymer matrix due to their covalent bonding. We should note that one disadvantage for the obtained polymer sensor is the delay in the manifestation of the sensor effect in the solid state compared to that in the solution. Therefore, future work should focus on eliminating this drawback.

## 4. Conclusions

We synthesized and characterized a novel 1,8-naphthalimide (NI3), structural isomer of the previously published 1,8-naphthalimide (NI4) [18], with sensing potential for protons and metal ions. Their spectral characteristics were compared and it was found that the position of the receptor fragment had a significant effect on the quantum yields of fluorescence. The effect of pH and various metal ions (Cu(II), Pb(II), Ni(II), Zn(II), Co(II) and Fe(III)) was investigated and it was found that the position of the receptor fragment (*N,N*-dimethylaminoethylamine) did not significantly impact proton detection. For the metal ions, good sensor ability was achieved when this fragment was introduced at the C-4 position. Computational tools were efficiently employed to delineate the relationship between the chemical structure and the optical and sensing properties of the novel 1,8-naphthalimide and its structural isomer (NI4). New copolymers with styrene were obtained (poly(St-co-NI3) and poly(St-co-NI4)) using 1,8-naphthalimide as a sensor fragment. Their spectral characteristics in a solid state and in a toluene solution were investigated. It was found that the copolymers emitted a yellow fluorescence. The copolymers were found to quench its fluorescent intensity after immersion in an aqueous solution that contained iron ions. The rest of the metal ions examined in this study did not affect the fluorescent intensity.

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
