# Peer review of "Spectral Characteristics and Sensor Ability of a New 1,8-Naphthalimide and Its Copolymer with Styrene"

_sensors, 2020, doi:10.3390/s20123501_

Round 1
Reviewer 1 Report
Staneva et al. describe the synthesis of an allyl-functionalized naphtalimide scaffold (NI3), to be exploited for pH and metal sensing. Its spectroscopic characteristics are experimentally and computationally investigated as a monomer and upon co-polymerization with styrene and compared with those of a regioisomer (NI4). The compound performance as proton sensor is comparable to that of NI4. On the contrary, light up response to metals is more selective, appearing in a more marked manner for iron and zinc, whereas NI4 produces considerable light up responses with all metals. However, this behaviour switches to a light-off upon incorporation into a polymeric scaffold.
The study is well conceived and complete, with most experimental aspects being soundly investigated and the observations supported by computational findings. The text is clear and the sequence of information is properly constructed and presented. Two main points raise my concern, though. First of all, the presented molecule is not as innovative in terms of structure. This would not be a problem if the small imparted changes led to a remarkable modification or improvement in its properties. However, this does not seem to be the case: the properties do not differ that much from those of the other monomer and, at the same time, the sensitivity for metals is still quite low (Cf. high LOD and LOQ values and high Ka value). The second point concerns the monomer incorporation in a polymer scaffold. The fact that this structural change leads to a light off response to metal complexation is not properly explained by the authors: why does this happen? At the same time, a light off response is way less interesting than a light-up response, since it is usually more difficult to detect and less selective in a complex environment. The final point on the polymer use concerns its recycling: the authors present it as a possible advantage, but do not provide data on the treatment to be applied for proper decomplexation and on the number of cycles that can be performed without losing in sensitivity.
Some other minor points:
1) NMR data for NI2 are missing
2) the carbon NMR of NI3 presents too many signals (18 chemically different carbons vs. 21 signals). Is the sample pure?
3) A representative plot of NI3 monomer absorption and fluorescence spectra would be a useful addition.
4) How is the quantum yield affected upon dissolution of the polymer in different solvents? Is it possible to acquire such data?
5) Has the chelation of Fe(III) by the terminal NMe2 and the oxygen of the carbonyl group been considered as a possible complexation mode?
Finally, the authors have inserted a remarkable amount of self-citations (12 out of 42). I believe this number should be tamed.
Author Response
Reviewer 1:
The study is well conceived and complete, with most experimental aspects being soundly investigated and the observations supported by computational findings. The text is clear and the sequence of information is properly constructed and presented. Two main points raise my concern, though.
First of all, the presented molecule is not as innovative in terms of structure. This would not be a problem if the small imparted changes led to a remarkable modification or improvement in its properties. However, this does not seem to be the case: the properties do not differ that much from those of the other monomer and, at the same time, the sensitivity for metals is still quite low (Cf. high LOD and LOQ values and high Ka value).
Authors’ reply: We agree with the reviewer that the molecule of NI3 does not have a complex structure, but our goal was to introduce a group that can copolymerise with traditional vinyl and acrylic monomers. In this case, the positions of these groups are reduced in terms of the place in the chromophore system. These derivatives, on the one hand, must be easily synthesized and chemically stable, and on the other hand, they must not change the spectral characteristics of 1,8-naphthalimide. Compound NI3 and its structural analogue NI4 allowed us to trace the structure-properties relationship.
The second point concerns the monomer incorporation in a polymer scaffold. The fact that this structural change leads to a light off response to metal complexation is not properly explained by the authors: why does this happen? At the same time, a light off response is way less interesting than a light-up response, since it is usually more difficult to detect and less selective in a complex environment.
Authors’ reply: We have introduced in the text an explanation of the quenching of the fluorescence and the possible formation of a complex with the carbonyl group, as a result of which the polarization changes and the fluorescence is quenched.
The final point on the polymer use concerns its recycling: the authors present it as a possible advantage, but do not provide data on the treatment to be applied for proper decomplexation and on the number of cycles that can be performed without losing in sensitivity.
Authors’ reply: An aqueous solution of EDTA was used to regenerate the iron ions from the polymer matrix. Six complete cycles of detection of metal ions were performed, during which a satisfactory behaviour of the fluorescent emission (compared to the original one) was observed. A possible explanation is given in the text.
NMR data for NI2 are missing
Authors’ reply: Added in the Experimental part.
2) the carbon NMR of NI3 presents too many signals (18 chemically different carbons vs. 21 signals). Is the sample pure?
Authors’ reply: Sorry for this mistake. It has been corrected.
3) A representative plot of NI3 monomer absorption and fluorescence spectra would be a useful addition.
Authors’ reply: Absorption and fluorescence spectra have been added to the main text (Figure 2).
4) How is the quantum yield affected upon dissolution of the polymer in different solvents? Is it possible to acquire such data?
Authors’ reply: We calculated the quantum yields of copolymer in toluene solution, the value is 0.51.
5) Has the chelation of Fe(III) by the terminal NMe2 and the oxygen of the carbonyl group been considered as a possible complexation mode?
Authors’ reply: Yes, this possible mode of complexation is presented in figure 13 of the revised manuscript. The result for the Gibbs free energy of this complex formation reaction (ΔG78=-31.2 kcal mol-1) indicates a spontaneous and energy-favorable complex formation process in water. The complex formation with the substituent at C-4 position is predicted to be also energy-favorable process (characterized by negative ΔG78 value, -21.8 kcal mol-1).
Finally, the authors have inserted a remarkable amount of self-citations (12 out of 42). I believe this number should be tamed.
Authors’ reply: The self-citations have been reduced.
Reviewer 2 Report
Thank you for submitting your manuscript to sensors. It is very well written and describes the use of novel polymers as pH and metal ion sensors. I would recommend this for publication after some minor issues have been addressed.
1) Can the authors compare how their new NI3, NI4 based pH and metal ion sensor compares to the state of the art? A table comparing the performance of this sensor to other polymer-based pH, metal ion sensors would be helpful to the sensors community to understand the impact of this work.
2) Can the authors comment on what the drawbacks of their sensor are? What kind of future work can be done to improve the application of these polymers as sensors and improve their sensor characteristics?
Author Response
Reviewer 2:
- Can the authors compare how their new NI3, NI4 based pH and metal ion sensor compares to the state of the art? A table comparing the performance of this sensor to other polymer-based pH, metal ion sensors would be helpful to the sensors community to understand the impact of this work.
Authors’ reply: There are few sources in the literature that treat the issue of optical sensory activity similar to the polymer sensors described in this paper, in which the sensor units are less than 1% of the base polymer monomer. Thus, it is difficult to make a comparison.
- Can the authors comment on what the drawbacks of their sensor are? What kind of future work can be done to improve the application of these polymers as sensors and improve their sensor characteristics?
Authors’ reply: This is a very good question. The polymer system could be improved by introducing various plasticizers that will facilitate the penetration of the analytes into the receptor fragments of 1,8-naphthalimides but this requires in-depth further research. Another way to improve the sensor characteristics would be to use another polymer matrix with better hydrophilic properties.
Reviewer 3 Report
I would suggest to consider the remarks presented in the attached pdf file prior acceptance of the manuscript.

Author Response
Reviewer 3:
The authors present a contribution to the filed of sensors based on their previous work https://www.hindawi.com/journals/jchem/2014/793721/, however the novelty of the work is not well justified. Therefore, I would suggest to consider the following remarks prior acceptance of the manuscript: I would kindly request that authors take care of the many typographical errors present in the text.
Authors’ reply: Corrected.
While, both a comma and a period are generally accepted decimal separators for international use, I would suggest that authors use either comma, or period.
Authors’ reply: Corrected.
The authors claim that the molecular weight values confirm the formation of high molecular weight polymers, whereas the values are considerably small: 0.97x10-5. Or is it possible that the authors meant 0.97x105 ? Please also mention the unit, which is g/mol.
Authors’ reply: Corrected.
It is claimed that the amount of chemically bonded 1,8-naphthalimide fluorophores has been determined by comparing the fluorescence intensity of the precipitated and unprecipitated polymer, the use of fluorescence spectroscopy being in this case preferable to the absorption as a more sensitive analytical method. However, I would ask the authors to consider NMR analysis to provide complementary data to confirm the chemically bonded 1,8-naphthalimide fluorophores.
Authors’ reply: Due to the low concentration of the NI3 linked to the main polystyrene macromolecule, it was not possible to use NMR spectroscopy to prove the chemical bonding between them. For this purpose, the new poly(St-co-NI3) copolymer has been characterized by gel permeation chromatography with double detection: refraction and UV-Vis at 423 nm. The values of the elution time obtained in both chromatograms are very close (Figure 1), which indicates that poly(St-co-NI3), absorb at 423 nm. This result indicates covalent bond formation between NI3 and main polystyrene chain.
The authors claim that the higher value of excitation maximum can be explained by the solid fixation of NI3 into the polymer matrix and the inability to conformational changes, which is also confirmed by the fluorescence maximum, nevertheless I would also request to add some literature reference which support this statement.
Authors’ reply: Done. Two references have been added (S. D. Zhao and T. M. Swager Sensory Responses in Solution vs Solid State: A Fluorescence Quenching Study of Poly(iptycenebutadiynylene), Macromolecules2005,38,9377-9384; Pinnock, S. S., Malele, C. N., Che, J., & Jones, W. E. (2011). The Role of Intermolecular Interactions in Solid State Fluorescent Conjugated Polymer Chemosensors. Journal of Fluorescence, 22(2), 583–589.)
Please plot the photophysical characteristics of copolymers in a Table by giving the exact concentration solutions.
Authors’ reply: Only one solvent has been used - 2% solution of the copolymer in toluene and the respective values of the spectral characteristics are given in the text.
Please discuss the relation between the Stokes shift and the solvent polarity interpreted on the basis of the Lippert Mataga equation and estimate the dipole moment change between the S1 and S0.
Authors’ reply: Figure 6 (“Dependence of Stokes shift (υA-υF ) on solvent polarity function Δf for NI3 in organic solvents: 1-toluene, 2-dichloromethane, 3-acetone, 4-DMF, 5-acetonitrile, 6- 2-propanol, 7-methanol”) and corresponding discussion have been added to the text.
The authors mention the good mechanical stability of the copolymers on Page 13, whereas no characterization is presented to confirm this statement. Please provide any analytical data (such as DMAT, TGA and DSC) which can confirm the mechanical and also thermal stability of the synthesized materials.
Authors’ reply: By the term "good mechanical stability of the copolymers…" we mean the properties of polystyrene. To the best of our knowledge and our experience in this field , the physico-mechanical properties of the polymers are not significantly affected by the presence of second monomer in such small amounts (in this case less than 1%). Difference between the copolymer and the homopolymer is not expected to be registered, so the suggested thermal tests are not performed.
Round 2
Reviewer 1 Report
The authors have satisfactorily replied to all my comments and implemented changes when needed. I therefore recommend the paper for publication in the present form.
Reviewer 3 Report
The current version of the manuscript is suitable to be accepted.